statistics/cognition/neuroscience

visual attention, parallel and serial processing, neural spike trains, hidden Markov model, correlated binomial model, statistical inference

**Author for correspondence:**
Susanne Ditlevsen
e-mail: susanne@math.ku.dk

# Distinguishing between parallel and serial processing in visual attention from neurobiological data

Kang Li[1,2], Mikiko Kadohisa[3], Makoto Kusunoki[3], John Duncan[3], Claus Bundesen[2] and Susanne Ditlevsen[1]

[1]Department of Mathematical Sciences, and [2]Department of Psychology, University of Copenhagen, Copenhagen, Denmark
[3]Department of Experimental Psychology, University of Oxford, Oxford, UK

KL, 0000-0001-8368-6930; SD, 0000-0002-1998-2783

Serial and parallel processing in visual search have been long debated in psychology, but the processing mechanism remains an open issue. Serial processing allows only one object at a time to be processed, whereas parallel processing assumes that various objects are processed simultaneously. Here, we present novel neural models for the two types of processing mechanisms based on analysis of simultaneously recorded spike trains using electrophysiological data from prefrontal cortex of rhesus monkeys while processing task-relevant visual displays. We combine mathematical models describing neuronal attention and point process models for spike trains. The same model can explain both serial and parallel processing by adopting different parameter regimes. We present statistical methods to distinguish between serial and parallel processing based on both maximum likelihood estimates and decoding the momentary focus of attention when two stimuli are presented simultaneously. Results show that both processing mechanisms are in play for the simultaneously recorded neurons, but neurons tend to follow parallel processing in the beginning after the onset of the stimulus pair, whereas they tend to serial processing later on.

## 1. Introduction

A fundamental question in theories of visual search is whether the process is serial or parallel for given types of stimulus material (for comprehensive reviews, see [1–3]). In serial search, only one stimulus is attended at a time, whereas in parallel search, several stimuli are attended at the same time. The question of serial versus parallel search has been extensively investigated by behavioural

methods in cognitive psychology, but it is still highly controversial. In this article, we briefly review extant empirical methods and their results and then present and exemplify a new method for distinguishing between serial and parallel visual search. The method is based on analysis of spike trains of simultaneously recorded single neurons measured in prefrontal cortex of rhesus monkeys while being exposed to a pair of stimuli, which the animal should detect and later respond to with a saccade towards a target object, first presented in [4]. A spike train is a sequence of recorded times at which a neuron fires an action potential, and it is believed that spike times are the primary way to transmit information in the nervous system. Point process modelling is a natural mathematical framework for addressing such phenomena, and we embed this into models of visual attention. In this article, we define *attention to a stimulus* on a cellular level as the stimulus a specific neuron is responding to. Thus, at a given time, attention is divided if all neurons are not responding to the same stimulus. This provides alternative means to quantify parallel versus serial visual processing. In particular, we do not average over trials or neurons, but model each spike train individually within a larger model, to allow the finer dynamics and interactions to reveal themselves. Thus, it is possible to follow attentional shifts in individual neurons, that would otherwise be averaged out if these do not happen simultaneously in the entire neuron population.

## 1.1. Behavioural methods for distinguishing between serial and parallel visual search

In typical experiments on visual search, the task of the observer is to indicate as quickly as possible if a certain type of target is present in a display. Positive (target present) and negative (target absent) mean response times are analysed as functions of the display set size $N$ (the number of items in the display). The method of analysis was laid out by Sternberg [5–7] and further developed by Schneider & Shiffrin [8]. The foundation is as follows.

In a simple serial model, the $N$ items are scanned one at a time. When an item is scanned, it is classified as a target or a distractor. The order in which items are scanned is independent of their status as targets versus distractors. A negative response is made when all items have been scanned and classified as distractors. Thus, the number of items processed before a negative response is made equals $N$. Furthermore, the rate of increase in mean negative response time as a function of $N$ equals the mean time taken to process one item, $\Delta t$. A positive response is made as soon as a target is found. Because the order in which items are scanned is independent of their status as targets or distractors, the number of items processed before a positive response is made varies at random between 1 and $N$ with a mean of $(1 + N)/2$. Thus, the rate of increase in mean positive response time as a function of $N$ equals $\Delta t/2$ (see [9] for experimental evidence of serial processing in a behavioural task).

In a parallel model of attention, several stimuli can be attended at the same time. The first detailed parallel model of visual processing of multi-element displays was the independent channels model proposed by Eriksen and his colleagues (e.g. [10,11]). It was based on the assumption that display items presented to separated foveal areas are processed in parallel and independently up to and including the stage of pattern recognition. The independent channels model has been used to account for effects of $N$ on error rates. The linear relations between mean response time and $N$ predicted by simple serial models are difficult to explain by parallel models with independent channels. However, the linear relations can be explained by parallel models with limited processing capacity [12,13]. Comparisons of serial and parallel models continue in the behavioural literature (e.g. [14–17]).

## 1.2. Method based on neurobiological data

As exemplified above, previous methods for distinguishing between serial and parallel visual search have been based on behavioural data, and the evidence obtained by these methods has been somewhat indirect. Moreover, these methods are all based on the assumption that processing is either serial or parallel, and that it stays the same throughout the trial. In this article, we present a new method for distinguishing between serial and parallel visual search, a method based on analysis of electrophysiological data. Furthermore, we propose measures to quantify the processing mechanism in a continuum between serial and parallel processing. The method relies on the probability-mixing model for single neuron processing [18,19], derived from the neural theory of visual attention [1,20], which states that when presented with a plurality of stimuli, a neuron only responds to one stimulus at any given time. The key property of a serial model would be that, at any instant, all neurons respond to the same stimulus, while in the parallel model, the neurons would be divided. These are the properties we intend to test in a simple case of search in a monkey. By probabilistic modelling and statistical inference using multiple simultaneously recorded spike trains, we infer and decode what the recorded neurons are responding to, providing a

way to distinguish between parallel processing and serial processing on a neuronal level. The new method appears more direct than previous methods, and it is possible to analyse the time evolution of the processing mechanism over the course of a trial.

Consider an experiment in which we record the action potentials or spikes from each of a number of neurons of the same type, e.g. a set of functionally similar neurons in visual cortex with overlapping receptive fields (e.g. [18]), or neurons in prefrontal cortex that are believed to be dynamically allocated to process task-relevant information (e.g. [4]), which are the neurons we analyse in this paper. Suppose two stimuli (stimulus 1 and 2) are both within the classical receptive fields of all of the recorded neurons, but otherwise the receptive fields are empty. In this situation, we may test whether processing is parallel, in the sense that some of the recorded neurons represent stimulus 1, while others represent stimulus 2 at any one instant. We may assume that a neuron represents stimulus 1 rather than stimulus 2 if the likelihood of the observed spike trains becomes higher by assuming that the neuron represents stimulus 1. We may also test whether processing is strictly serial by testing, for example, whether there is a time interval $\Delta_1$ in which all of the neurons represent stimulus 1 and a time interval $\Delta_2$, non-overlapping with $\Delta_1$, in which all of the neurons represent stimulus 2. Strictly parallel or strictly serial processing of two or more stimuli may hardly be expected in a biological system, and must be regarded as idealizations. However, we will show how to measure the goodness of approximation of search processes in the brain to simple serial and parallel search, as well as the time evolution of the processing mechanism.

In the following sections, we first present the statistical methods and probabilistic models that we employ to distinguish between parallel and serial processing, and then show and discuss the results obtained using both simulated data and experimental data. In §2, we introduce the experimental data used to measure the degree of parallel and serial processing in a realistic biological situation, and explain our proposed statistical criteria and measures to distinguish between parallel and serial processing. We propose two models, the hidden Markov model (HMM) and the correlated binomial model (CBM), to account for the spike train data in an attention framework, and calculate their likelihood functions. The maximum-likelihood estimates (MLEs) provide a prior measurement of parallel versus serial processing. We can also decode the momentary focus of attention given the fitted models, which provides a posterior measurement. In §3, we present the results of the analysis conducted on the experimental data. As the sequel shows, we found evidence of parallel processing (different neurons responding to different stimuli) early in trials but serial (focused) processing (all neurons responding to the same stimulus) later in trials.

# 2. Material and methods

We present two models that relate the theories of visual attention to neuronal behaviour, providing a tool to distinguish or quantify between parallel and serial processing through spike train analysis. Under the assumption of serial processing, the neurons are correlated, acting together as a population. This dependence can arise through two different pathways: (i) there exists an underlying variable driving the neurons towards attending to the same stimulus, creating a dependence, even if the neurons are conditionally independent given the state of this underlying variable; and (ii) the neurons are directly positively correlated, driving them to synchronize their attention.

The first pathway is naturally described by an HMM, where the hidden Markov chain switches between different states influencing the neuronal attention. If time is discretized and there are two stimuli, this leads to a mixture of binomials at each discretized time step, where the number of components in the mixture distribution equals the number of states of the Markov chain. The binomial distributions provide probabilities of the number of neurons attending to each stimulus, in dependence of the hidden state of the Markov chain. The second pathway can be represented by a CBM, a mixture of an ordinary binomial and a modified Bernoulli [21], which is used independently at each discretized time step. For both models, the attended stimulus for each neuron is unobserved, and the inference is based on spike train data. We estimate parameters using MLE by marginalizing out the unobserved attention variables. The estimated parameters in either model describe neuronal properties and are used to obtain a *prior* measurement of the degree of parallel or serial processing. For both models, we also decode the hidden states from the posterior probabilities of the latent attention variables, i.e. an estimate of the stimulus the neurons were most probably attending to given their observed spike trains. The decoding of attentional behaviour provides a *posterior* measurement of the degree of parallel or serial processing. The diagram in figure 1 summarizes the flow of the analysis including parameter estimation, decoding and interpretation.

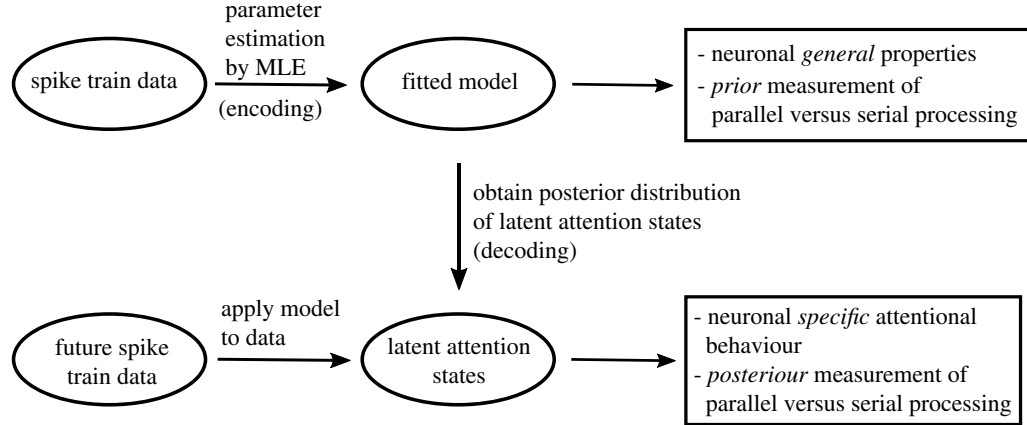

**Figure 1.** Flow diagram of the analysis.

## 2.1. Experimental data

The neural spike train data are recorded from neurons in prefrontal cortex of two rhesus monkeys presented with two visual stimuli [4]. They studied dynamic attentional construction, and found that in the early stage after stimulus onset when processing competing stimuli, the global attention is distributed among all objects, with each neuron having a tendency towards its contralateral hemifield. In the late stage, the global attention is reallocated and neurons are redirected to the target stimulus. The data contain multiple simultaneously recorded neurons responding to two competing stimuli. Note that neurons were not selected for task-related responses; instead neuronal activity was isolated before starting the task, and all recorded neurons were included in the analysis. The data are organized in daily sessions, and each session consists of a different set of recorded neurons. At the beginning of a session and before starting the task, microelectrodes were advanced until neuronal activity could be isolated. Each neuron was then included in all trials up to where it was lost. Then the following trials were with the remaining neurons. We only analyse the sessions where at least five neurons are recorded to have enough data to distinguish between parallel and serial processing, yielding a total of 48 sessions. Note, however, that at single trials towards the end of a session, there might be less than five neurons if many neurons got lost during the course of a session. Otherwise, there has been no preselection of data. The monkey fixed attention on a central red dot on a computer screen, then each trial began with a central cue indicating the target object of the specific trial. Each of two cues was paired with one of the two alternative targets. After a brief delay, a choice display was presented for 500 ms containing two objects: one to the right and one to the left of the fixation point. The two stimulus objects consisted of a combination of either the cued target (T), an inconsistent non-target (NI) because it was used as a target on other trials, a consistent non-target (NC) never serving as a target, or nothing but a grey dot (NO). The stimulus locations were denoted by whether they were contra- or ipsilateral with respect to the recorded neuron. After a brief delay, the monkey was rewarded with a drop of liquid for a saccade to the T location if a T had been shown, or if no T had been presented, for maintaining fixation (no-go response) for later reward. In the following, we call a combination of two stimuli a *condition*. Table 1 shows the 12 possible conditions. More details on the data can be found in the electronic supplementary material.

We analyse the spike trains of the choice phase where the two stimuli are shown. To account for neuronal response times, we discard the first 100 ms after stimulus onset, using the interval from 100 to 500 ms in the choice phase when estimating the parameters of the two models. In figure 2 are shown the recorded spike trains of an example cell during this phase and 100 ms before and after. The neuron seems to favour the target T with a higher firing rate, and its attention starts from the contralateral stimulus and is later redirected to the target stimulus, following the overall tendency of most neurons reported by Kadohisa *et al.* [4].

## 2.2. Measures for parallel versus serial processing

Here we define different measures of the degree of serial and parallel processing based on the estimated parameters of the models when a population of $n$ neurons are presented with two non-overlapping

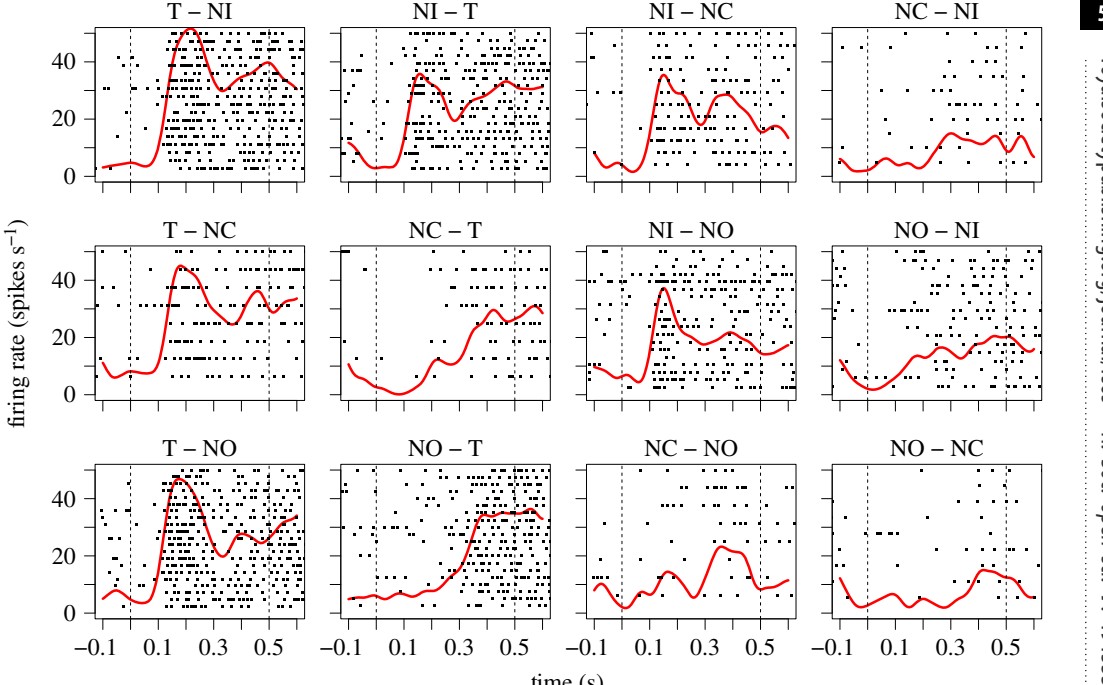

**Figure 2.** Raster plots of measured spike trains recorded from an example cell (MN110411task_3_0). Kernel smoothing estimates of the firing rates are shown in red. The dashed lines indicate the interval of the choice phase where two stimuli are shown. The 12 conditions (table 1) are indicated in the title of the subplot. The left title indicates the stimulus on the contralateral side, and the right indicates the stimulus on the ipsilateral side with respect to the recorded neuron.

**Table 1.** The 12 conditions used in the trials (combinations of stimuli). Conditions can be merged into three groups: target in the contralateral side (conditions 1–3), target in the ipsilateral side (conditions 4–6) and all combinations with no target (conditions 7–12). Contra- and ipsilateral sides are with respect to the recorded neuron. T, target; NI, inconsistent non-target; NC, consistent non-target; NO, no display.

| condition | 1 | 2 | 3 | 4 | 5 | 6 | 7 | 8 | 9 | 10 | 11 | 12 |
|---|---|---|---|---|---|---|---|---|---|---|---|---|
| contralateral | T | T | T | NI | NC | NO | NI | NI | NC | NC | NO | NO |
| ipsilateral | NI | NC | NO | T | T | T | NC | NO | NO | NI | NI | NC |

stimuli in their receptive fields. These measures will vary with time, i.e. depend on the time since stimulus onset, but for ease of notation, we suppress time from the notation here. Later we will introduce the time dependency. We assume a homogeneous situation where all neurons follow the same distribution and are exchangeable, except for individual firing rates as responses to single stimuli. These measures are based on the basic probability-mixing model for the attention of single neurons employed in [18,19], where a neuron responds to a stimulus mixture with certain probabilities, such that the single neuron at any given time represents only one of the stimuli in the mixture. First, we consider the marginal distribution of the attended stimulus for each neuron. Let $p$ denote the marginal probability of attending to one of the stimuli, say stimulus 1, such that the probability of attending stimulus 2 is $1 - p$, where $0 < p < 1$. If the neurons are independent, then the probability that all neurons attend the same stimulus is $p^n + (1 - p)^n$, and if the neurons are positively correlated, this is a lower bound of the probability that all neurons attend the same stimulus. Thus, $p$ provides a measure of the tendency of serial or parallel processing. A narrow distribution (extreme probability, $p$ either close to 0 or 1) favours serial processing, since in this case most neurons will attend the same stimulus. A wide distribution (non-extreme probability, $p$ close to 0.5) favours parallel processing, since in this case neuronal attention will tend to split between the two stimuli. Second, we consider correlations between neurons. Since the neurons are exchangeable, the correlation coefficient,

**Table 2.** Effects of neural attentional probability and correlation to serial and parallel processing.

| | extreme probability $p \approx 0$ or $p \approx 1$ | non-extreme probability $p \approx 1/2$ |
|---|---|---|
| strong correlation $\rho \approx 1$ | serial | serial |
| weak correlation $|\rho| \ll 1$ | serial | parallel |

denoted by $\rho$, between any two neurons (pairwise correlation) is identical. Stronger positive correlation implies more tendency to serial processing, no matter what $p$ is. Thus, if either the correlation is strong ($\rho$ close to 1) or $p$ is close to 0 or 1, serial processing is favoured, while if both the correlation is weak and the probability is not extreme, parallel processing is favoured. We summarize the different cases in table 2.

We propose a single statistic as an alternative measure to distinguish between serial and parallel processing that take into account the entire joint distribution. Assume a stimulus mixture of two components and a population of $n$ neurons reacting to the mixture. The number of neurons, $Z$, attending the first stimulus follows a distribution with probability mass function (PMF) $f(z)$ for $z \in \{0, 1, \ldots, n\}$, such that $P(Z = z) = f(z)$, which depends on the specific model. A distribution centred around $n/2$ indicates apparent parallel processing, and a distribution centred at 0 and/or $n$ indicates apparent serial processing. Note that this distribution incorporates both the marginal probability of attention of the single neurons and the correlation between neurons. Define the statistic $D_n$ by

$$D_n = \frac{\sum_{z=0}^{n} |z - n/2| f(z)}{n/2}. \tag{2.1}$$

The statistic $D_n$ can be explained as a normalized expected deviation between the number of neurons attending to one stimulus and the half of the total number of neurons. If we split the neuron population according to which stimulus they attend giving two proportions (summing to 1), then $D_n$ is the average difference between the two proportions, and it can take values between 0 and 1. The smaller $D_n$ is, the more parallel processing is favoured. The $D_n$ statistic depends on the total number of recorded neurons $n$. However, if we consider specific models for the PMF, for example, the binomial models introduced below, the dependence of $n$ can be removed by using the asymptotic version

$$D^* = \lim_{n \to \infty} D_n, \tag{2.2}$$

which provides a measure for the entire neuronal population relevant for the given task.

The comparison of serial and parallel processing catches the differences among simultaneously recorded neurons within one trial in terms of their attended stimulus, which is hard or impossible to analyse by traditional methods by averaging across neurons and trials. We thus develop a new methodology modelling each single spike train and the correlation between spike trains. The serial and parallel processing can be distinguished using the estimated parameters. To summarize, to measure the degree of serial/parallel processing, we use the attentional probability $p$, the correlation of neuronal attention $\rho$, and the deviation statistics $D_n$ or $D^*$.

## 2.3. Models

In this section, we present two models to explain the spike train data in an attention framework. We discretize the 400 ms of the trial where both stimuli are presented, and which we use for the analysis, into $I$ smaller intervals and let the models evolve dynamically over these intervals. Within any of these small time intervals, we assume that the attention of each neuron is not changing. Within a trial, let $X_t^i \in \{0, 1\}$ denote the attended stimulus of neuron $i$ at time $t$ for $i = 1, \ldots, n$, $t = 1, \ldots, I$, and let $Y_t^i$ denote the spike train of neuron $i$ in the $t'$th interval. We set $X_t^i = 1$ when neuron $i$ attends stimulus 1 at time $t$, and $X_t^i = 0$ when attending stimulus 2. Stimulus 1 is defined as the contralateral stimulus with respect to the recorded neuron, stimulus 2 is the ipsilateral stimulus. Thus, $p_t = P(X_t^i = 1)$. This probability depends on the stimulus pair; however, at $t = 1$ it is only related to the location of the attended stimulus, since this is the initiation of the processing mechanism before the specific stimuli are perceived, and is thus the same for all stimulus pairs.

## 2.3.1. Point process model for the spike trains

We assume a point process model for the spike trains and perform maximum-likelihood estimation [18,22,23]. The conditional intensity function (CIF) of a general point process model is defined by

$$h(t|H_t) = \lim_{\Delta t \to 0} \frac{\Pr(N(t + \Delta t) - N(t) = 1|H_t)}{\Delta t}, \tag{2.3}$$

where $N(t)$ is the number of spikes in the interval $(0, t]$, and $H_t$ denotes the spike history up to time $t$. Then $h(t|H_t)\Delta t$ approximates the probability of observing a spike in $(t, t + \Delta t)$ for $\Delta t$ small.

Suppose a spike train $y$ in the interval $[T_s, T_e]$ contains the spike times $y = \{t_1, t_2, \ldots\}$ with $T_s \leq t_1 < t_2 < \cdots \leq T_e$, and that it attends the same stimulus during the entire interval. The probability of observing $y$ given the attended stimulus $x_t$ is given by [22,23]

$$P(y|x_t) = \left[\prod_{\tau \in y} h(\tau|H_\tau; x_t)\right] \exp\left\{-\int_{T_s}^{T_e} h(s|H_s; x_t)\,\mathrm{d}s\right\}, \tag{2.4}$$

where $h(s\,|\,H_s; x_t)$ is the CIF in equation (2.3), which we model using

$$h(s|H_s; x_t) = r \exp\left\{\sum_{j=1}^{10} \beta_j \Delta N_{s-ju}\right\}. \tag{2.5}$$

The base firing rate $r := r^i(x)$, where $i = 1, \ldots, n$ indicates the neuron, is neuron specific and a function of the attended stimulus $x$ and the location (contra- or ipsilateral). Note that only the attended stimulus is relevant, not the condition. For each neuron, there are therefore seven rate parameters, representing T, NI and NC at either side, and a parameter for NO. The exponential term models the influence of the past spikes during the previous 10 ms on the neuronal activity. The constant $u = 1$ ms is the discretization unit determined by the experiment, and $\Delta N_t$ denotes whether there is a spike ($\Delta N_t = 1$) or not ($\Delta N_t = 0$) in the time interval $[t, t + u)$. For simplicity, we assume that only past spikes of the neuron itself have an effect. All neurons are assumed to share the same set of parameters $\beta_j$, $j = 1, 2, \ldots, 10$.

Let $\mathcal{M}$ denote the considered conditions (stimulus pairs) and let $|\mathcal{M}|$ denote the number of conditions. For simplicity, we do not always distinguish between all 12 conditions shown in table 1, but sometimes merge them into classes, such that there will be fewer parameters to estimate. In particular, we will consider the three classes of conditions indicated in table 1, defined by whether there is a target in the stimulus pair, and if there is, whether it is contra- or ipsilateral. Under condition $m$, let the set $\mathcal{K}_m$ contain all the conducted trials. In trial $k$, let the set $\mathcal{N}_k$ contain all the simultaneously recorded neurons and let $y_t^{\mathcal{N}_k}$ denote the spike trains from these neurons in the $t'$th interval, and likewise for the hidden attentional states $X_t^{\mathcal{N}_k}$. Each $\mathcal{N}_k$ is a subset of the set of all neurons $\mathcal{N}$ used in the session, $\mathcal{N}_k \subseteq \mathcal{N}$, because not all neurons are used in all trials (electronic supplementary material, figure S1).

## 2.3.2. Hidden Markov model and a mixture of binomial distributions

To combine the visual attention hypotheses with neuronal dynamics, we adopt an HMM. The HMM assumes some underlying unobserved variable that drives the attention of the neurons. The HMM is defined over the $I$ time steps. We let the probabilities $p$ of the single neurons, which can be interpreted as attentional weights, depend on the state of the underlying HMM, which introduces correlation between neurons, even if they are conditionally independent given the hidden state, and the probabilities evolve over time following the dynamics of the HMM. Note that this implies that within each of the $I$ intervals, model parameters governing the stochastic neuronal activity (the spike train generation) are constant. We use three hidden states, which describe three attentional regimes. These could, for example, be attention mainly directed to the contralateral side, attention mainly directed to the ipsilateral side, or approximately equal attention to both sides. Note, however, that the probabilistic features within states are data driven. A transition between hidden states introduces a weight reassignment of the attention to the stimuli, and thus, new laws for the generation of spike trains. Let $C_t \in \{1, 2, 3\}$ denote the hidden state at time $t$. Figure 3 shows a diagram of the HMM for $I = 3$. Conditional on $C_t$, the $\{X_t^i\}_{i=1,\ldots,n}$ are independent.

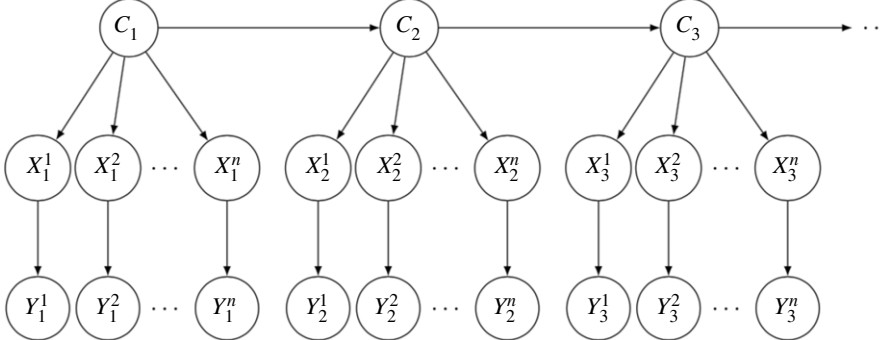

$C_t \in \{1, 2, 3\}$: hidden state during time interval $t$

$X_t^i \in \{0, 1\}$: attended stimulus during time interval $t$ for neuron $i$

$Y_t^i$: spike train in interval $t$ for neuron $i$

**Figure 3.** Diagram of the hidden Markov model. The HMM and attentional states from a group of $n$ neurons to a mixture of two stimuli, using $l = 3$ discretized time steps.

Let the initial distribution of the Markov chain be given by $\lambda$ and the transition probability matrix (TPM) by $\Gamma$:

$$\left. \begin{aligned} \lambda &= [\lambda_1 \quad \lambda_2 \quad \lambda_3] \\ \text{and} \qquad \Gamma &= \begin{bmatrix} \gamma_{11} & \gamma_{12} & \gamma_{13} \\ \gamma_{21} & \gamma_{22} & \gamma_{23} \\ \gamma_{31} & \gamma_{32} & \gamma_{33} \end{bmatrix}, \end{aligned} \right\} \tag{2.6}$$

where $\sum_{k=1}^{3} \lambda_k = 1$, $\sum_{l=1}^{3} \gamma_{kl} = 1$ for $k = 1, 2, 3$ and $\lambda_k, \gamma_{kl} \geq 0$ for $k, l = 1, 2, 3$. Here, $\lambda_k = P(C_1 = k)$ and $\gamma_{kl} = P(C_{t+1} = l \,|\, C_t = k)$ for $t > 1$. Let the vector $\pi_t := \lambda \Gamma^{t-1}$ denote the distribution of $C_t$, thus, $\pi_{t,k} = P(C_t = k)$. The TPM $\Gamma$ depends on the stimulus pair, but the initial distribution $\lambda$ is only related to the location of the attended stimulus. We denote by $\Gamma_m$ the TPM of condition $m$.

Conditional on $C_t$, neurons are assumed independent. Denote the probability of attending stimulus 1 given state $c$ by $\alpha_c = P(X_t^i = 1 | C_t = c)$, yielding the matrix:

$$A = \begin{bmatrix} \alpha_1 & 1 - \alpha_1 \\ \alpha_2 & 1 - \alpha_2 \\ \alpha_3 & 1 - \alpha_3 \end{bmatrix}. \tag{2.7}$$

### 2.3.2.1. Attention probabilities and correlations

The vector

$$P_t = \lambda \Gamma^{t-1} A = \pi_t A = (P(X_t^i = 1), P(X_t^i = 0)) = (p_t, 1 - p_t) \tag{2.8}$$

contains the probabilities of attention to the two stimuli. Straightforward calculations yield the moments and the correlation $\rho_t$ between $X_t^i$ and $X_t^j$

$$\mathbb{E}(X_t^i) = \lambda \Gamma^{t-1} [\alpha_1, \alpha_2, \alpha_3]' = p_t, \tag{2.9}$$

$$\text{Var}(X_t^i) = p_t(1 - p_t), \tag{2.10}$$

$$\mathbb{E}(X_t^i X_t^j) = \lambda \Gamma^{t-1} [\alpha_1^2, \alpha_2^2, \alpha_3^2]', \tag{2.11}$$

$$\text{Cov}(X_t^i, X_t^j) = \mathbb{E}(X_t^i X_t^j) - \mathbb{E}(X_t^i)\mathbb{E}(X_t^j) \tag{2.12}$$

$$\text{and} \quad \rho_t = \frac{\text{Cov}(X_t^i, X_t^j)}{\text{Var}(X_t^i)}, \tag{2.13}$$

where $'$ denotes transposition. The values $p_t$ and $\rho_t$ can be used to measure the degree of serial and parallel processing as indicated in table 2.

### 2.3.2.2. A mixture of three binomials

By marginalizing out the hidden state $C_t$, the HMM structure implies that at each time point $t$ the neuronal attention of the $n$ neurons follows a mixture of three binomial distributions, $\text{Bin3}(\pi_t, \alpha, n)$. Here, $\alpha = (\alpha_1, \alpha_2, \alpha_3)$ are the probability parameters of the three binomials, and the weights are given by $\pi_t$. The number of binomial trials equals the number of simultaneously recorded neurons $n$. The

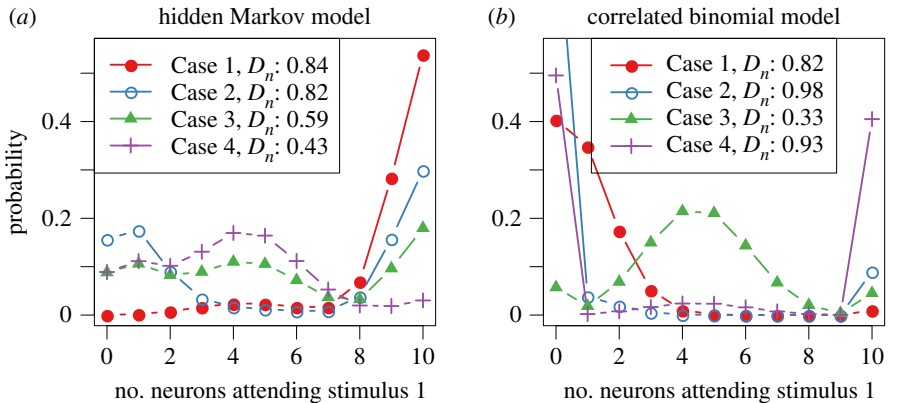

**Figure 4.** The probability mass function of the number of neurons attending stimulus 1. Parameter values are given in table 3. The statistic $D_n$ is defined in equation (2.1) and measures the degree of parallel/serial processing. Higher values indicate serial processing. (*a*) HMM and (*b*) CBM.

**Table 3.** Parameters, probabilities of attention, correlations and the deviation statistics $D_{10}$ and $D^*$ for the HMM and the CBM. Values used to produce figure 4.

| | $\pi_{t,1}$ | $\pi_{t,2}$ | $\pi_{t,3}$ | $p$ | $\rho$ | $D_{10}$ | $D^*$ |
|---|---|---|---|---|---|---|---|
| | hidden Markov model, $\alpha = (0.95, 0.45, 0.1)$ | | | | | | |
| Case 1 | 0.9 | 0.1 | 0.0 | 0.9 | 0.25 | 0.84 | 0.82 |
| Case 2 | 0.5 | 0.05 | 0.45 | 0.54 | 0.69 | 0.82 | 0.82 |
| Case 3 | 0.3 | 0.45 | 0.25 | 0.51 | 0.41 | 0.59 | 0.52 |
| Case 4 | 0.05 | 0.7 | 0.25 | 0.39 | 0.17 | 0.43 | 0.32 |
| | correlated binomial model | | | | | | |
| Case 1 | – | – | – | 0.1 | 0.1 | 0.82 | 0.82 |
| Case 2 | – | – | – | 0.1 | 0.9 | 0.98 | 0.98 |
| Case 3 | – | – | – | 0.45 | 0.1 | 0.33 | 0.19 |
| Case 4 | – | – | – | 0.45 | 0.9 | 0.93 | 0.91 |

PMF for the mixture of three binomials is

$$f_{\text{Bin3}}(z|\pi_t, \alpha, n) = \pi_{t,1}f(z|\alpha_1, n) + \pi_{t,2}f(z|\alpha_2, n) + (1 - \pi_{t,1} - \pi_{t,2})f(z|\alpha_3, n), \tag{2.14}$$

where $f(z|\alpha_k, n) = \binom{n}{z}\alpha_k^z(1 - \alpha_k)^{n-z}$ is the PMF of the binomial distribution.

The $D_n$ statistic is calculated using equation (2.1). For the mixture of three binomials in (2.14), the asymptotic version is given by

$$D^* = \lim_{n\to\infty} D_n = 2(\pi_{t,1}|\alpha_1 - 0.5| + \pi_{t,2}|\alpha_2 - 0.5| + (1 - \pi_{t,1} - \pi_{t,2})|\alpha_3 - 0.5|). \tag{2.15}$$

Figure 4*a* illustrates how the probability and the correlation affect serial and parallel processing for the HMM using $n = 10$ neurons for four different parameter settings, Cases 1–4. The four cases show increasing degree of parallel processing. The parameter settings are shown in table 3, together with the derived values of the probabilities of attention, correlations and deviation statistics $D_{10}$ and $D^*$.

### 2.3.2.3. Likelihood function

We denote the conditional probability of the $\mathcal{N}_k$ spike trains at time $t$ given $C_t$ by a diagonal matrix:

$$P(y_t^{\mathcal{N}_k}|C_t) = \begin{bmatrix} P(y_t^{\mathcal{N}_k}|C_t = 1) & 0 & 0 \\ 0 & P(y_t^{\mathcal{N}_k}|C_t = 2) & 0 \\ 0 & 0 & P(y_t^{\mathcal{N}_k}|C_t = 3) \end{bmatrix}. \tag{2.16}$$

**Table 4.** Parameters and interpretation of the HMM and the CBM. Parameters to be estimated in each session. Interpretation of and differences between the models.

| parameter | explanation | dimension |
|---|---|---|
| **hidden Markov model** | | |
| $\lambda$ (equation (2.6)) | initial distribution, the same for all conditions $\mathcal{M}$ | 2 |
| $\Gamma_m$ (equation (2.6)) | transition probability matrix for each condition $m \in \mathcal{M}$ | $6|\mathcal{M}|$ |
| $A$ (equation (2.7)) | conditional probability of neuronal attention | 3 |
| **correlated binomial model** | | |
| $\rho_{t,m}$ (equation (2.19)) | correlation coefficients for $m \in \mathcal{M}$ and $t = 1, \dots, l$ | $|\mathcal{M}| \cdot (l-1) + 1$ |
| $p_{t,m}$ (equation (2.19)) | probability parameter for $m \in \mathcal{M}$ and $t = 1, \dots, l$ | $|\mathcal{M}| \cdot (l-1) + 1$ |
| **common to both models** | | |
| $r^i$ (equation (2.5)) | base firing rates, one for each neuron in $\mathcal{N}$ | $7|\mathcal{N}|$ |
| $\beta$ (equation (2.5)) | weights in the CIF model, the same for all neurons in $\mathcal{N}$ | 10 |

| interpretation | HMM | CBM |
|---|---|---|
| motivation | extends the probability-mixing model with dynamic weight reassignment | treats neuronal attention as correlated binomial variables |
| neuronal correlation within a time interval | modelled through the hidden state of the Markov chain | modelled directly by parameters |
| neuronal correlation between time intervals | modelled by the Markov chain | independent |
| parameter dimension | $15 + 6|\mathcal{M}| + 7|\mathcal{N}|$ | $12 + 2|\mathcal{M}|(l-1) + 7|\mathcal{N}|$ |

The likelihood function of all spike trains in one session is then given by

$$L = \prod_{m \in \mathcal{M}} \prod_{k \in \mathcal{K}_m} \left\{ \lambda P(y_1^{\mathcal{N}_k}|C_1) \prod_{t=2}^{T} [\Gamma_m P(y_t^{\mathcal{N}_k}|C_t)] \right\}. \tag{2.17}$$

By conditioning on the hidden attentional states $X_t^{\mathcal{N}_k}$, we obtain

$$\begin{aligned}
P(y_t^{\mathcal{N}_k}|C_t) &= \prod_{i \in \mathcal{N}_k} P(y_t^i|C_t) \\
&= \prod_{i \in \mathcal{N}_k} [P(X_t^i = 1|C_t)P(y_t^i|X_t^i = 1) + P(X_t^i = 0|C_t)P(y_t^i|X_t^i = 0)] \\
&= \prod_{i \in \mathcal{N}_k} [\alpha_{C_t} P(y_t^i|X_t^i = 1) + (1 - \alpha_{C_t})P(y_t^i|X_t^i = 0)],
\end{aligned} \tag{2.18}$$

where $P(y_t^i|x_t^i)$ is given in equation (2.4). We obtain MLEs of the parameters by maximizing the likelihood function. The parameters to be inferred are summarized in table 4.

### 2.3.3. Correlated binomial model

In the CBM, the neurons are assumed directly correlated. It was studied in [21,24] and is denoted by $\mathrm{CBin}(n, p, \rho)$. In this model, the number of neurons $z$ attending stimulus 1 follows a mixture of two distributions. One is an ordinary binomial distribution with parameters $n$ and $p$. The other is a fully correlated distribution where $z \in \{0, n\}$, which can be viewed as a modified Bernoulli distribution with

support $\{0, n\}$ with parameter $p$. The weight of the Bernoulli component is the correlation coefficient $\rho$. 
The probability mass function is given by

$$f_{\mathrm{CBin}}(z|n, p, \rho) = (1 - \rho)f(z|n, p) + \rho p^{(z/n)}(1 - p)^{(n-z/n)}I_{\{0,n\}}(z), \tag{2.19}$$

where $I_{\{0,n\}}(z)$ is the indicator function which equals 1 for $z \in \{0, n\}$ and 0 otherwise.

The distribution of the neuronal attentions at $t = 1$ is assumed identical for all stimulus pairs, $\mathrm{CBin}(n, p_1, \rho_1)$, and at $t > 1$ they follow $\mathrm{CBin}(n, p_{t,m}, \rho_{t,m})$ for stimulus pair $m$. Contrary to the HMM, the behaviour at different time steps is independent. Instead, the correlation is modelled directly by the parameter $\rho$. Compared with the HMM, where the correlation is caused by the attentional reassignment according to a hidden state, the CBM is more direct.

The probability of attention is given by the parameter $p_{t,m}$, and the correlation is $\rho_{t,m}$. The asymptotic version of the deviation statistic $D^*$ defined in equation (2.2) to measure the degree of serial and parallel processing is given by

$$D^* = 2(1 - \rho)|p - 0.5| + \rho. \tag{2.20}$$

Figure 4b shows the PMF of the correlated binomial distribution for the four parameter settings given in table 3.

### 2.3.3.1. Likelihood function

Under the CBM, the attention of the simultaneously recorded neurons follow a mixture of a binomial and a modified Bernoulli. The likelihood of the spike trains in condition $m$ at time $t$ in trial $k$, $y_t^{\mathcal{N}_k}$, is given by

$$P_m(y_t^{\mathcal{N}_k}) = (1 - \rho_{t,m}) \underbrace{\prod_{i \in \mathcal{N}_k} [P(y_t^i|X_t^i = 1)p_{t,m} + P(y_t^i|X_t^i = 0)(1 - p_{t,m})]}_{\text{binomial}}$$

$$+ \rho_{t,m} \underbrace{\left\{ p_{t,m} \prod_{i \in \mathcal{N}_k} P(y_t^i|X_t^i = 1) + (1 - p_{t,m}) \prod_{i \in \mathcal{N}_k} P(y_t^i|X_t^i = 0) \right\}}_{\text{modified Bernoulli}}, \tag{2.21}$$

where $P(y_t^i|x_t^i)$ is given in equation (2.4). The likelihood of the data of an entire session is

$$L = \prod_{m \in \mathcal{M}} \prod_{k \in \mathcal{K}_m} \prod_{t=1}^{T} P_m(y_t^{\mathcal{N}_k}). \tag{2.22}$$

We summarize the differences of the HMM and the CBM in table 4. In both models, it is assumed that in the early stage, i.e. the first discretized interval from 100 ms to $100 + 400/T$ ms, neuronal attention is only affected by the position of stimuli (ipsi- or contralateral) and not by stimulus types (T, NI, NC or NO). This assumption is supported by the empirical findings by firing rate averaging showing attentional reallocation over time [4]. It is also assumed that under the same stimulus types, the attentional parameters are identical, implying that in all the trials of one condition, neurons follow the same distribution and differences from trial to trial due to randomness.

## 2.4. Decoding the attentional state

Decoding means to infer the attended stimulus from the observations and the estimated parameters. The basic idea is to compute the posterior distribution given the spike train, $P(X_t^i|Y = y)$, for neurons $i = 1, \dots, n$, which provides estimates of the attended stimulus of single neurons. The estimated PMF of the number of neurons attending one stimulus, $P(\sum_i^n X_t^i|Y = y)$, is then used to calculate $D_n$, defined in equation (2.1). In the following, the decoding is explained for the two models in more detail. To show the main idea, we suppress time and neuron indicator from the notation for the moment, denoting the hidden state by $C$, the attended stimulus by $X$ and the spike train data by $Y$. The posterior of $X$ given $Y = y$ is

$$P(X|Y = y) = \sum_c P(X|C = c, Y = y)P(C = c|Y = y). \tag{2.23}$$

The strategy is to first estimate $P(C = c|Y = y)$ and then $P(X|C = c, Y = y)$ conditional on $C = c$. We are particularly interested in the PMF and the deviation statistic of the attended stimuli, which we can calculate using $P(X|C = c, Y = y)$ for different states $C$.

## 2.4.1. Decoding in the hidden Markov model

The hidden states $C_t$ in the HMM model is decoded at each discretized time step by the forward–backward algorithm. Let $y_{s:t}^{\mathcal{N}_k}$ denote the spike trains in intervals $s$ to $t$, for $1 \leq s < t \leq I$ in trial $k$, where $\mathcal{N}_k$ denotes the simultaneous recorded neurons in the $k$'th trial. The probability of $C_t$ conditional on the observed spike trains at all time intervals 1 to $I$ can be expressed as

$$P(C_t|y_{1:I}^{\mathcal{N}_k}) \propto P(y_{t+1:I}^{\mathcal{N}_k}|C_t)P(C_t|y_{1:t}^{\mathcal{N}_k}), \tag{2.24}$$

where

$$P(C_t|y_{1:t}^{\mathcal{N}_k}) \propto P(y_t^{\mathcal{N}_k}|C_t) \sum_{c_{t-1}} P(C_t|c_{t-1})P(c_{t-1}|y_{1:t-1}^{\mathcal{N}_k}) \tag{2.25}$$

is the forward probability, calculated recursively by a forward sweep over 1 to $I$, and

$$P(y_{t+1:I}^{\mathcal{N}_k}|C_t) = \sum_{c_{t+1}} P(y_{t+2:T}^{\mathcal{N}_k}|c_{t+1})P(y_{t+1}^{\mathcal{N}_k}|c_{t+1})P(c_{t+1}|C_t) \tag{2.26}$$

is the backward probability, calculated recursively by a backward sweep over 1 to $I$. When calculating the forward and backward probabilities, the likelihood conditional on the hidden state, $P(y_t^{\mathcal{N}_k}|C_t)$, is obtained by conditioning on the neuronal attention $\{x_t^i\}_{i \in \{\mathcal{N}_k\}}$:

$$P(y_t^{\mathcal{N}_k}|C_t) = \prod_{i \in \mathcal{N}_k} \sum_{x_t^i \in \{0,1\}} P(y_t^i|x_t^i)P(x_t^i|C_t). \tag{2.27}$$

After decoding the hidden state $P(C_t|y_{1:I}^{\mathcal{N}_k})$, the next task is to decode $\{X_t^i\}_{i \in \{\mathcal{N}_k\}}$ conditional on $C_t$:

$$P(x_t^i|y_t^{\mathcal{N}_k}, C_t) = P(x_t^i|y_t^i, C_t) \propto P(y_t^i|x_t^i, C_t)P(x_t^i|C_t). \tag{2.28}$$

For all spike trains in trial $k$, $y_{1:I}^{\mathcal{N}_k}$, we have thus obtained the discrete posterior distributions of the hidden states $P(C_t|y_{1:I}^{\mathcal{N}_k})$ and the attended stimulus of each spike train $P(X_t^i|y_t^i, C_t)$, at all time steps $t = 1, \dots, I$. This yields the marginal posterior $P(X_t^i|y_{1:I}^{\mathcal{N}_k}) = \sum_{C_t \in \{1,2,3\}} P(X_t^i|y_t^i, C_t)P(C_t|y_{1:I}^{\mathcal{N}_k})$.

At each time step $t$, conditional on $C_t$, spike trains are independent and the posterior probabilities $P(X_t^i|y_t^i, C_t)$ are different from spike train to spike train. Thus, the attended stimuli of all neurons follow a Poisson binomial distribution, a generalization of the ordinary binomial distribution where each Bernoulli trial has a distinct success probability [25]. The PMF of the Poisson binomial distribution is calculated numerically using methods from [26]. Marginalizing out $C_t$, at each time step $t$ we then have a mixture of three Poisson binomial distributions. The PMF of this mixture distribution can be regarded as probabilities of the number of neurons that have attended stimulus 1, conditional on their observed spike trains. The deviation statistic $D_n$ defined in equation (2.1) can then be obtained from the PMF.

## 2.4.2. Decoding in the correlated binomial model

In the CBM, spike trains between different time steps and different trials are independent (except for the memory component, the exponential term in equation (2.5)). Thus, decoding can simply be done independently for each discretized time step in each trial. Now, let $C_t$ be an index indicating either the binomial or the Bernoulli component in the mixture. As previously, we first decode $C_t$ by calculating $P(C_t|y_t^{\mathcal{N}_k})$, then find the PMF by calculating $P(X_t^i|y_t^i, C_t)$. We have

$$P(C_t|y_t^{\mathcal{N}_k}) \propto P(y_t^{\mathcal{N}_k}|C_t)P(C_t), \tag{2.29}$$

where the two cases $C_t = 1$ and $C_t = 2$ are given by the two components in equation (2.21). Then for each case of $C_t$ we decode the attended stimulus $X_t^i$. When $C_t = 1$, i.e. the binomial case, $X_t^i$ is obtained for each spike train independently with $P(x_t^i|y_t^i, C_t = 1) \propto P(y_t^i|x_t^i, C_t = 1)P(x_t^i|C_t = 1)$, resulting in a Poisson binomial distribution. When $C_t = 2$, i.e. the fully correlated Bernoulli case, the attended stimuli of all neurons are the same, which is obtained by $P(x_t|y_i^{\mathcal{N}_k}, C_t = 2) \propto P(y_i^{\mathcal{N}_k}|x_t, C_t = 2)P(x_t|C_t = 2)$, and the result is still a modified Bernoulli. Finally, the PMF is a mixture of a Poisson binomial and a modified Bernoulli.

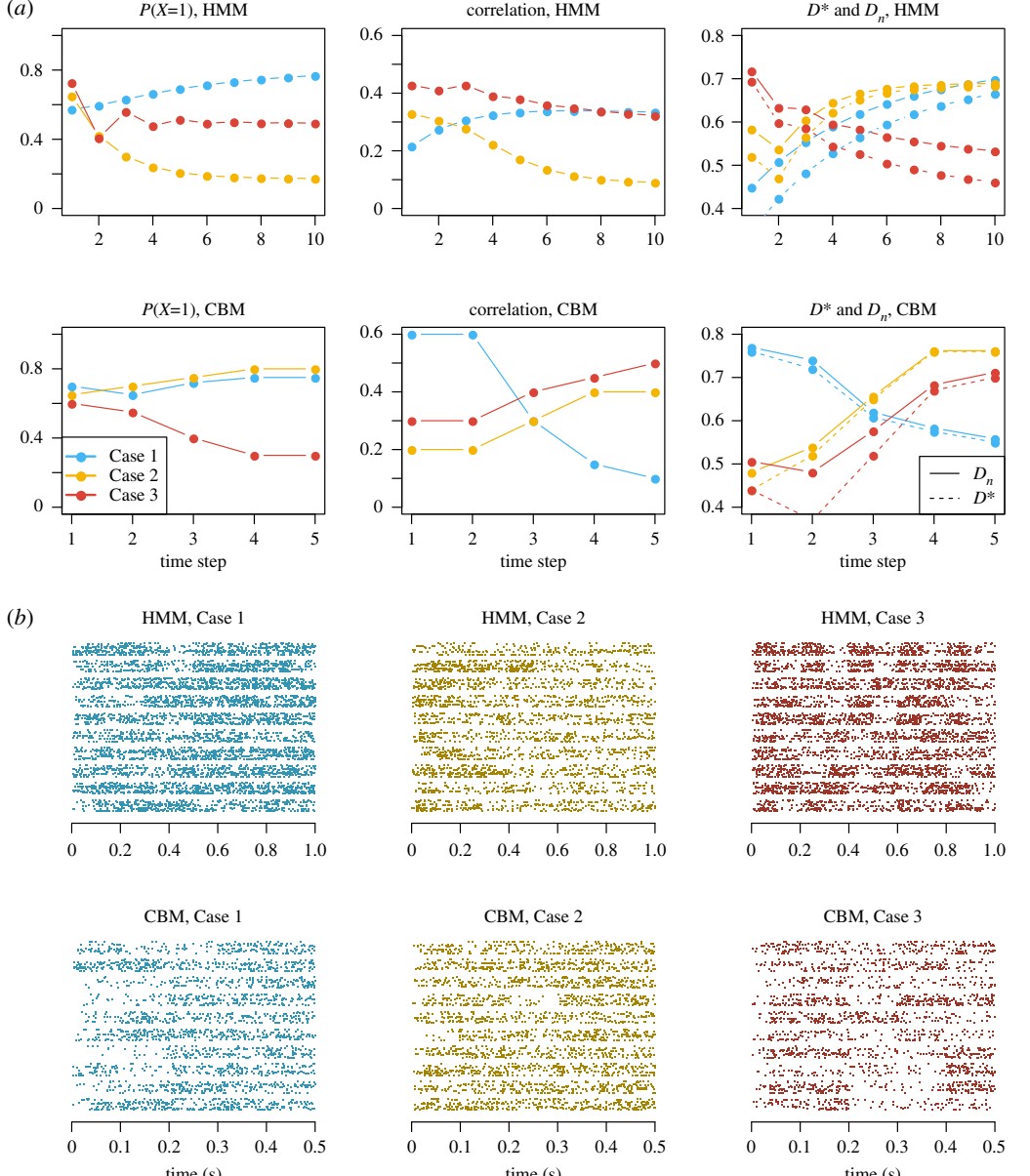

**Figure 5.** Examples of simulated data for the HMM and the CBM. (*a*) The probabilities of attending the contralateral stimulus, *p* (left), the pairwise correlation coefficients $\rho$ (middle) and the deviation statistic values $D_n$ and $D^*$ (right) are shown as functions of time. Different colours represent the three parameter settings. (*b*) Example spike trains are shown for the corresponding case and model. In each sub-figure, 10 trials are shown, separated by horizontal white space lines. In each trial, 10 simultaneous spike trains are plotted.

# 3. Results

## 3.1. Simulated data

We first simulate spike train data and check if our models and methods work properly on the simulated data. For both the HMM and the CBM, we consider three parameter settings. In all cases, we use 10 simultaneously recorded neurons, repeated for 20 trials. The base rates and response weights are the same for the three cases. We consider only one stimulus condition, such that each neuron has two base rate parameters, one for the contralateral and one for the ipsilateral sides. For the HMM, we use a time step of 0.1 s and a total of 10 time steps. For the CBM, we use a time step of 0.1 s and a total of five time steps. Note that the number of parameters in the CBM scales linearly with $I|\mathcal{M}|$ (table 4), and it is therefore not statistically viable to discretize with a large $I$ for the CBM. This is not the case for the HMM. Figure 5 shows probabilities of attending the contralateral stimulus, the pairwise correlation coefficients, and the deviation statistic values as functions of time for different parameter settings, as well as example simulated spike trains.

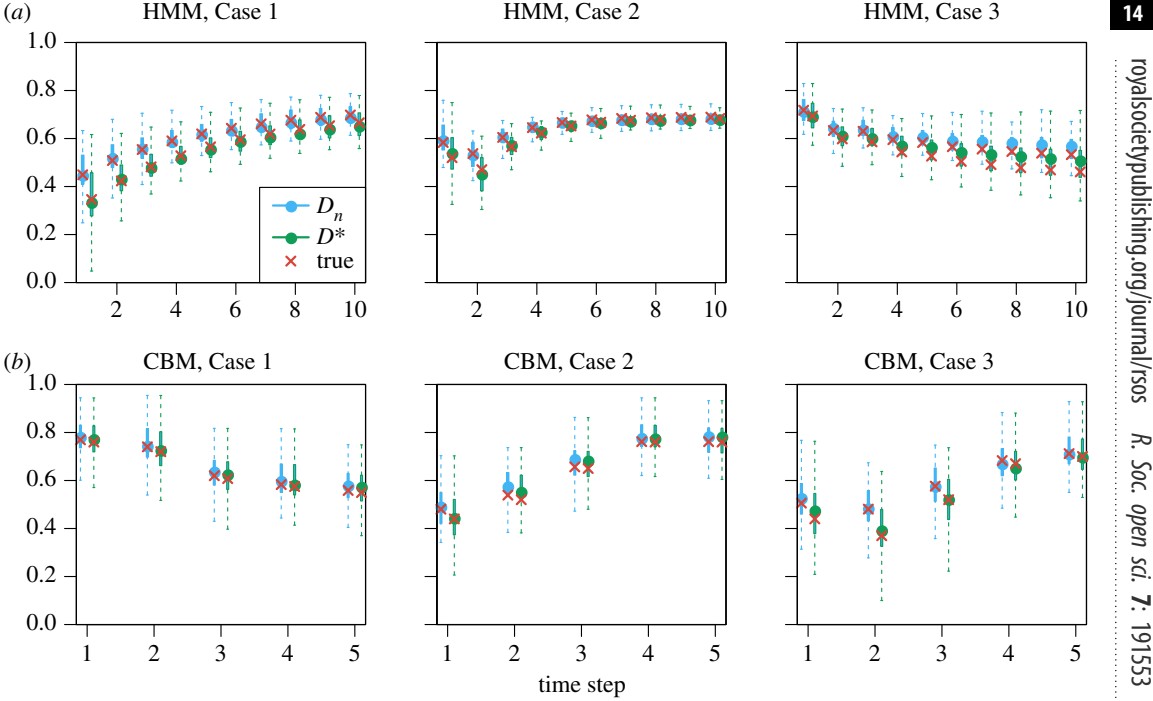

**Figure 6.** Simulation study: deviation statistics values computed from parameter estimates and true parameters. The estimates of $D_n$ (equation (2.1)) are shown in blue, and the estimates of $D^*$ (equation (2.2)) are shown in green as quantiles of the 100 repetitions. The dashed lines represent the full 0–100% quantiles, and the solid lines represent the 25–75% quantiles. The blue dots are the medians. The red crosses are the true values used in the simulation. (a) HMM and (b) CBM.

We apply the model fitting to the simulated data, and the simulation and model fitting procedure are repeated 100 times. Figure 6 shows the distribution of the estimates of the deviation measures. Further details, results and figures can be found in the electronic supplementary material.

The conclusion from this simulation study is that parameters can be successfully estimated, and the $D_n$ and $D^*$ values computed from the parameter estimates are close to the true values. The $D_n$ values from the decoding analysis have large variances, due to the small sample size of 10 neurons. However, the median $D_n$ values from the 100 decoding repetitions are often close to the encoding results based on the parameter estimates.

## 3.2. Experimental data

The experimental spike train data from [4] were fitted to both models. For a discretization with $I$ steps, an equal length of $400/I$ ms were assigned to all time intervals. Three different discretizations of $I = 3$, 5 or 10 were used, and two different classes of conditions with either all 12 or only 3 classes determined by whether there is a target in the stimulus pair, and in that case, whether it is contra- or ipsilateral (table 1). However, for the CBM only $I = 3$ or 5 were used since otherwise there are too many parameters to estimate, especially if all 12 conditions are individually modelled. The models were fitted to each of the 48 sessions independently.

### 3.2.1. Parameter estimation in HMM

Figure 7 illustrates parameter estimates for the HMM under different condition and step number settings. Figure 7a shows the probability of attending the stimulus at the contralateral side, $p_t = P(X_t = 1)$ at different time steps and types of conditions, as kernel density plots from all 48 estimates. It shows that neuronal attention slightly prefers the contralateral stimulus in the beginning right after stimulus onset (the black density curve is centred towards larger values than 0.5), and later on tends to follow T and avoid NO. Note that here we conduct model inference using all 12 conditions, and only combine similar conditions for presentation.

In figure 7b, the estimates of the correlation $\rho_t$ are plotted against the estimates $|p_t - 0.5|$ for each time step $t = 1, \ldots, 5$, on top of a two-dimensional kernel density estimate (bandwidth: 0.25) of the points as

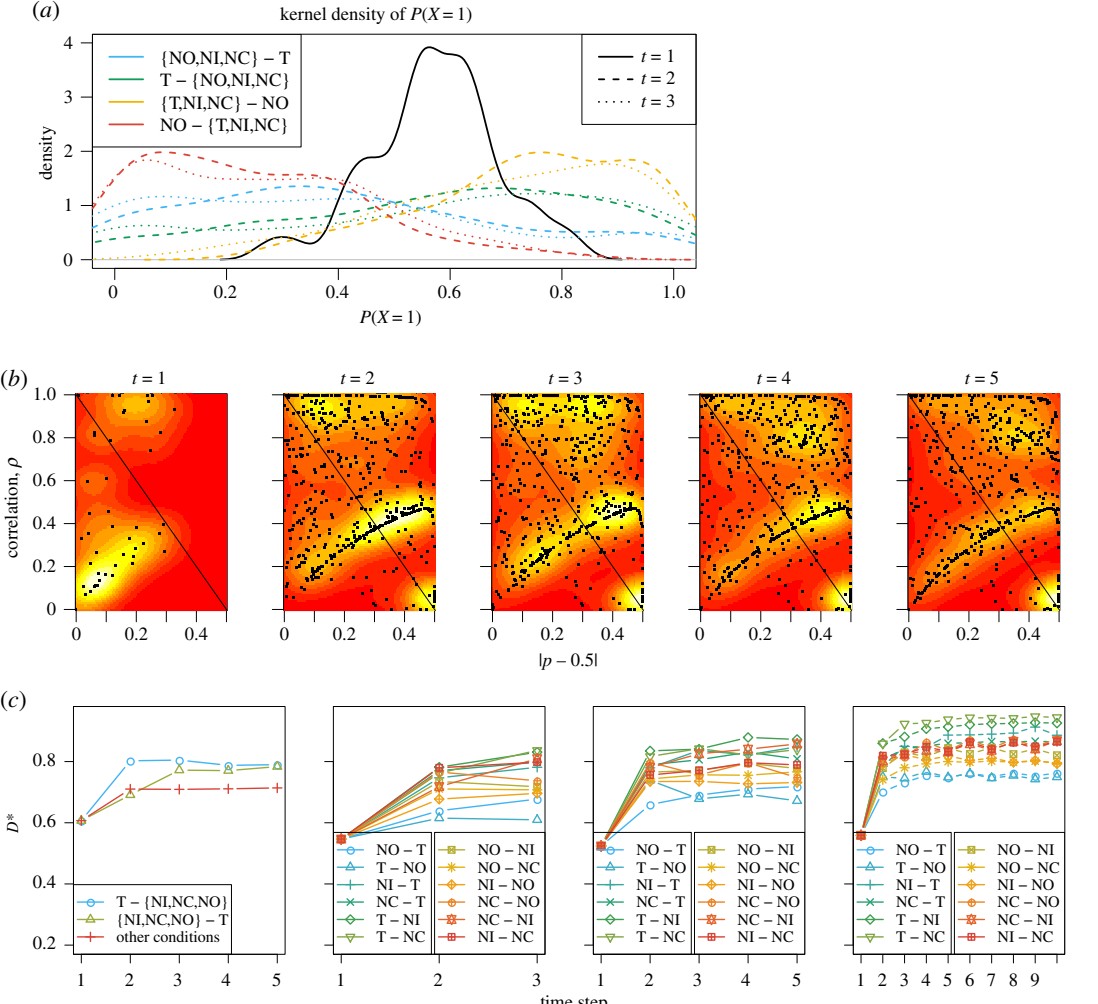

**Figure 7.** Experimental data: results for the HMM. (*a*) Kernel density representation of the estimates of $p_t$, the probability of attending to the contralateral stimulus, obtained using $l = 3$ and all 12 conditions (combined for presentation). At $t = 1$, all conditions follow the same distribution, so there is a single black curve. For the subsequent time steps, the condition types are: stimulus pairs with T on the ipsilateral side; stimulus pairs with T on the contralateral side; stimulus pairs with NO on the ipsilateral side; and stimulus pairs with NO on the contralateral side. (*b*) Estimates of correlation $\rho$ against estimates of probability extremeness $|p - 0.5|$ at the different time steps obtained using $l = 5$ and all 12 conditions, on top of a two-dimensional kernel density estimate as heatmaps serving as a visual tool presenting smoothed estimates of the tendency of parallel/serial processing from the point estimates. Yellow implies higher point density, red implies lower point density. (*c*) Estimates of $D^*$ (equation (2.2)) using three merged conditions with $l = 5$ (left), all 12 conditions with $l = 3$ (middle left), $l = 5$ (middle right), and $l = 10$ (right).

heatmaps. The heatmaps serve as a visual tool presenting smoothed estimates of the tendency of parallel/serial processing from the point estimates, as measured by the probability $p$ and the correlation $\rho$ as indicated in table 2. There are 48 estimates in the leftmost panel at $t = 1$ (no difference between conditions), and $48 \times 12$ estimates in the remaining panels from 12 conditions in 48 sessions. A straight line is plotted on the anti-diagonal for easier reading. The lower left region of the heatmap represents a tendency of parallel processing, and all other regions represent a tendency of serial processing. In the leftmost panel corresponding to the first time step, a big portion of the estimates fall in the lower left region. This implies parallel stimulus processing in the early stage. Later, the estimates tend to move to the right and upper regions, indicating serial processing. However, there are points on both sides of the straight line at all time steps. This is evidence supporting both processing mechanisms at all time steps throughout the entire spike train.

In figure 7*c*, we investigate the asymptotic deviation statistic $D^*$. The average $D^*$ is calculated over the 48 session estimates for each condition, for different discretizations and merging of conditions. In all cases, $D^*$ increases over time, implying stronger serial processing.

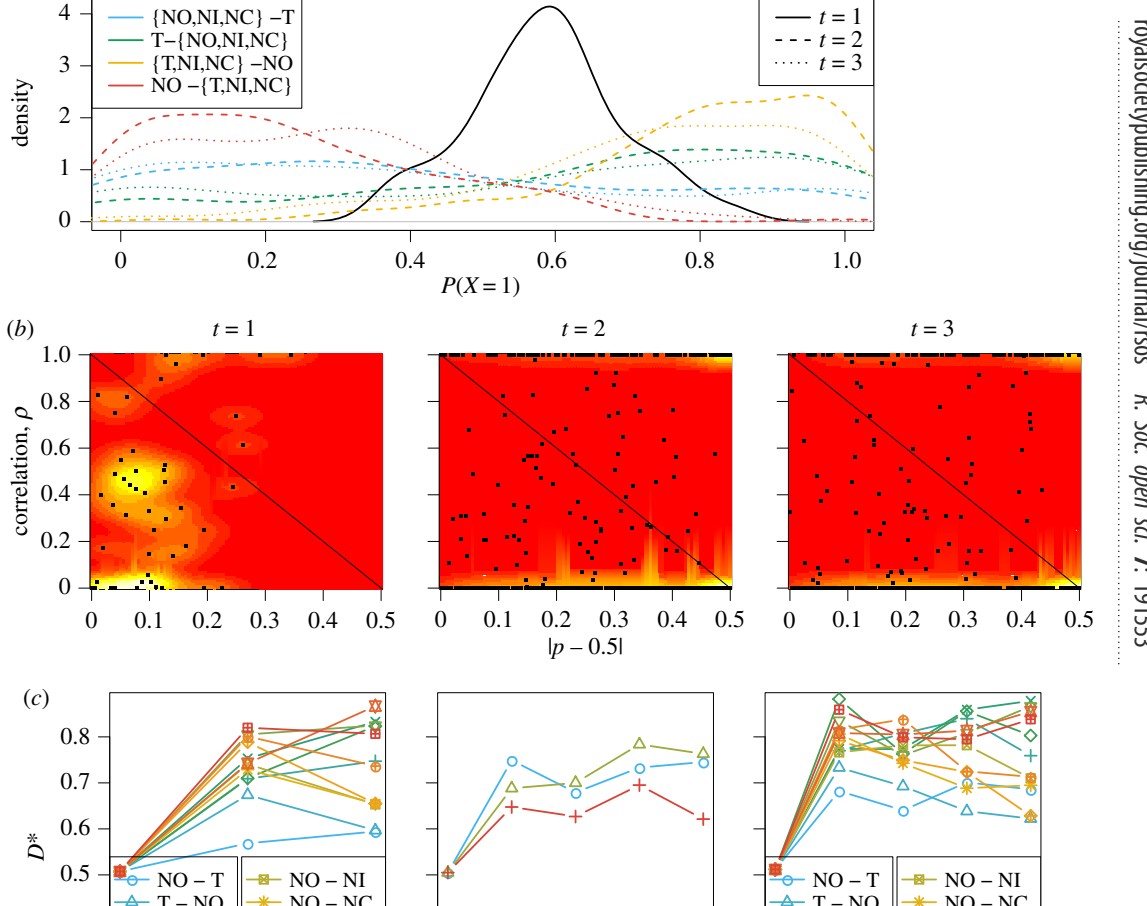

**Figure 8.** Experimental data: results for the CBM. Panels (*a,b*) use all 12 conditions with *l* = 3. Panel (*c*) uses 12 conditions with *l* = 3 (left), 3 merged conditions with *l* = 5 (middle), and 12 conditions with *l* = 5 (right). See caption of figure 7 for explanation.

### 3.2.2. Parameter estimation in CBM

The estimates of the CBM are shown in figure 8. The results are similar to the HMM. Figure 8*b* shows apparent parallel processing at $t = 1$, while later on the estimates of the correlation and the probability becomes more extreme. For $t > 1$, most of the estimates of the correlation coefficient are close to either 1 or 0, meaning one component in equation (2.19) is dominating over the other. This is because of the small number of simultaneously recorded neurons in most trials (typically two to four neurons), which is insufficient for obtaining good estimates in a mixture model. This is a weakness of the CBM since it only contains two extreme components representing either full independence or full correlation. To check this hypothesis, we looked at the estimates from session 'MN110411' with the largest average number of simultaneously recorded neurons (the rightmost neuron in the electronic supplementary material, figure S1b), and found that the estimates of the correlation lie almost uniformly across 0–1, indicating that the estimates of either 0 or 1 of the correlation in other sessions can be an artefact of small sample sizes.

### 3.2.3. Decoding

Here, we decode the attended stimulus of the neurons conditional on the observed spike trains. The parameters used in the decoding algorithms are the estimated parameters obtained by MLE.

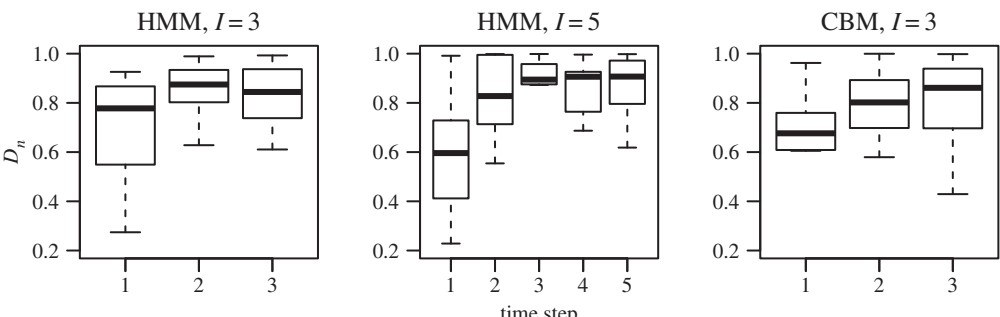

**Figure 9.** Decoding of an example trial using different models. The target is located in the ipsilateral side. The top panels show the results of the HMM model using $I = 3$ and 5. The bottom panel shows the CBM with $I = 3$. (*a*) Ten simultaneously recorded spike trains from a trial in session 'MN110411'. Note that the same data are presented three times. The dashed lines indicate the discretization. (*b*) The posterior probability of each spike train attending the contralateral stimulus at each time step. Red colour indicates higher probability of attending the contralateral stimulus and blue colour indicates higher probability of attending the ipsilateral stimulus. (*c*) The posterior probability of the hidden state $C$ in the HMM responsible for top-down control (not relevant for the CBM). (*d*) The PMF of the number of neurons attending to the contralateral stimulus conditional on the spike train data for each time step, with the $D_n$ values shown in the legend. These values are calculated by equation (2.1) using the estimated PMFs.

**Figure 10.** Decoding results of $D_n$. Only trials with more than 10 simultaneously recorded neurons are included. Results are from the HMM using $I = 3$ and 5, and the CBM using $I = 3$, respectively, indicated by the titles of the figures.

Figure 9 shows the decoding of the attended stimulus for an example trial containing 10 simultaneously recorded spike trains in session 'MN110411', condition NO-T. The same dataset is decoded using the HMM with $I = 3$ and 5, and the CBM with $I = 3$. The values of the $D_n$ statistics are calculated based on the decoded probabilities of a single trial containing simultaneously recorded neurons.

In figure 10, we show box-plots of the $D_n$ values from multiple trials across all sessions as a function of time: HMM with $I = 3$ and 5, and CBM with $I = 3$. Note that the results in figures 7*c* and 8*c* are prior measures based on estimated parameters, and figure 10 shows posterior measures based on the decoded

attended stimulus for specific spike train data. If a trial has too few simultaneously recorded spike trains, the $D_n$ values calculated from the decoded attentions will have large variance across trials, and results are unreliable. Therefore, we only consider trials with at least 10 simultaneously recorded spike trains. Note that this is only in the decoding analysis; in the parameter estimation all trials were used. We see a similar trend as in the $D_n$ values based on the estimates: the $D_n$ values are increasing with time, indicating stronger serial processing. Finally, in both models and at all time steps, there is evidence of both parallel and serial processing, implied by the wide box-plots.

## 4. Discussion

In this study, we combine the point process spiking neuron models describing spike trains with the neural interpretations of serial and parallel processing hypotheses in visual search. We propose an HMM and a CBM to describe neuronal attention in neurophysiological measurements from prefrontal cortex in rhesus monkeys. Results show that parallel processing is favoured in some sessions while serial processing is favoured in other sessions, and there is evidence for both parallel and serial processing at all time steps. Overall, we see a tendency towards parallel processing in the early stage after stimulus onset, and serial processing in the late stage. This means that, right after stimulus onset, the neurons tend to split their attention to different stimuli, and later the neurons become more synchronized, sharing the same attended stimulus. Furthermore, at the early stage neurons prefer the contralateral stimulus, while in the late stage neurons favour the T and avoid NO, which agrees with the study conducted by averaging across spike trains [4]. Although we do not know the special functions of the tested cells, the data strongly suggest that, as time goes by, prefrontal cortex moves from a state in which a multitude of inputs drive the response, to a state dominated by one of the stimuli, also in trials with no target.

The early state of parallel processing can be related to feedforward or bottom-up processing, where the sensory inputs are being processed before higher level cognitive modulatory influences of recurrent feedback or top-down processing have begun [27,28]. In the later stage, where top-down signals have had time to modulate the attention, the neural activity tends to synchronize around the attended object, resembling serial processing. Thus, cognitive modulatory influences guiding attentional effects in recurrent feedback connections occur after a small delay, and are related to serial processing, where all processing capacities are being directed towards the attended object. Similar results have been observed in event-related potentials in electroencephalography (EEG) measurements [29]. They found that forward connections are sufficient to explain the data in early periods after stimulus onset, whereas backward connections become essential after around 220 ms. Even if the exact timing of the switch between bottom-up and top-down signals is not clear, there is evidence that after 200 ms back-projections play a prominent role, even if selective responses are elicited already 100 ms after stimulus onset (see [28] and references therein). Quantification of the relative contribution of feedforward and feedback signals characterizing visual perception remains unclear, and thus, the concepts of parallel and serial processing and our suggested analysis tools provide a useful mean for elucidating these questions.

Our empirical findings support the selective attention for identification model (SAIM) [30,31]. The SAIM models the human ability to perform translation-invariant object identification in multiple object scenes. SAIM suggests that central for this ability is an interaction between parallel competitive processes in a selection stage and an object identification stage. Presented with two objects, the selection process begins with attending two stimuli, corresponding to parallel processing, and then as time passes only one stimulus is represented, corresponding to serial processing. Qualitatively, this behaviour is in agreement with our empirical findings. Another interesting parallel between SAIM and the HMM presented here is that the underlying variable $C_t$ driving the neuronal attention in the HMM could correspond to the selection network in SAIM which directs the focus of attention towards a stimuli.

Decoding analysis provides posterior probabilities of neuronal attentions, yielding an estimate of the PMF and therefore also of $D_n$. This can be used to analyse attentional behaviour for any given simultaneously recorded spike trains in future trials, as well as tracking the dynamics of the specific stimuli single neurons are attending. The conclusions regarding parallel and serial processing from the overall distribution of $D_n$ on all trials and sessions from the decoding analysis are the same as in the prior analysis using parameter estimates. Note that although both the prior and posterior analyses provide similar results, the conclusions regarding neuronal attentional properties should be drawn from the prior analysis based on the MLE. The MLE gives the optimal estimation of the neuronal

properties based on all the available data. The decoding analysis, on the other hand, estimates what the neuron's attention could have been during a specific trial based on the data from this trial, and the uncertainty of the decoding is represented by posterior distributions.

In [4], parallel processing in the early stage was reported. The same conclusion is drawn from our analysis, where we find that the neurons prefer the contralateral stimulus in the early stage, and integrating both hemispheres gives simultaneous parallel processing. Furthermore, there exists not only such parallel processing considering the whole brain, but also parallel processing based on neurons in a single recording site, as supported by our finding. Though the simultaneously recorded neurons in one location show a tendency towards the contralateral stimulus in the early stage, there is strong evidence showing they split their attention between stimuli located on both sides in a parallel way.

The models here are fitted to the specific dataset from [4] and the model structure contains the experimental conditions specific for this dataset. However, with trivial adjustments, the models also apply to generic neurophysiological data that consist of simultaneously recorded spike trains. Currently, the models and methods only support two stimuli, and a future extension is the generalization to an arbitrary number of stimuli.

The two models, the HMM and the CBM, yield slightly different results regarding the degree of serial and parallel processing. This is partly because the two models are based on different assumptions. In the HMM, the neuronal attention is guided by a common control, but otherwise the neurons are independent. In the CBM, the neurons affect each other directly. In the HMM, there is temporal correlation through the transition matrix, whereas in the CBM, the dynamics in a time interval is independent of the past (except for the memory component in the spiking model). The models have a different number of parameters, and consequently, different flexibility to fit the data at a cost of statistical power. The number of parameters in the CBM increases linearly with number of discretization steps $I$, and it is thus not possible to make a fine discretization in this model. However, for small $I$, the CBM has fewer parameters than the HMM, and thus more statistical power, especially if there are many conditions. The biological reality of attention, which we try to describe with these simple models, is complicated, and the two models approximate the reality and explain neural attention from different perspectives. Furthermore, the experimental data are noisy with limited sample size and the models contain a large number of parameters, which leads to large variance of estimators. Even if the difference between the two models is large in a given trial or session, the overall results of the two models over a large number of sessions produce similar conclusions. However, specific comparisons should be made under the same model, such as comparing the processing mechanisms under different conditions.

An essential assumption in our models is that each neuron attends to only one stimulus at any given time, which is supported by some recent studies [18,19]. However, the assumption might not be valid in general. Suppose instead that a neuron responds to two stimuli with a firing rate that is a weighted average of the firing rates it would have when presented with only one of the stimuli. Then our models can be considered as approximations, where the probability $p$ corresponds to the weight on stimulus 1. In that case, the statistic $D_n$ in equation (2.1) still makes sense, where $f(z)$ is then a distribution on the entire real interval $[0, N]$, and not a distribution on the integers from 0 to $N$.

We assume that in the early stage, neuronal attention is only affected by the position of stimuli (ipsi- or contralateral) and not by stimulus types (T, NI, NC or NO). To test this assumption, we also ran the HMM using three time steps, allowing each condition to have its own initial probabilities. The conclusions remain the same, processing is more parallel right after stimulus onset, and becomes more serial later on (results not shown). Because of the increased number of parameters, the CBM with condition-specific initial probabilities is not statistically viable. Another issue is the variability between sessions for the same model. Assuming the whole prefrontal area follows a probabilistic model, we aim at estimating the model parameters of the entire system. However, in each session only a small subset of 5–11 simultaneously recorded neurons is available, and the number is even smaller for single trials (electronic supplementary material, figure S1), with each neuron having its distinct firing rate and attentional pattern (figure 2 and electronic supplementary material, figure S2). Thus, there is a large variance of the estimates from session to session, and a more stable result is obtained by averaging and applying kernel density estimation. To obtain more accurate estimates, it is necessary to have a larger number of simultaneously recorded neurons.

Ethics. All experimental procedures were approved by the UK Home Office and were in compliance with the guidelines of the European Community for the care and use of laboratory animals (EUVD, European Union directive 86/609/EEC).

Data accessibility. The data and the computer code are uploaded as part of the electronic supplementary material.
Authors' contributions. K.L., C.B and S.D. conceived the research, drafted the paper and interpreted the results. K.L. wrote the computer code, did the analysis and produced the figures. K.L. and S.D. wrote the final version of the paper. Mi.Ka. Ma.Ku. and J.D. managed and carried out the experimental work. All authors read and approved the final version of the paper.
Competing interests. The authors declare they have no competing interests.
Funding. The work was part of the Dynamical Systems Interdisciplinary Network, University of Copenhagen Excellence Programme for Interdisciplinary Research (PI: S.D.). J.D. received funding from Medical Research Council, Research Councils UK (grant no. SUAG/002/RG91365).

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
