## [Reviewer comments · Royal Society Open Science]

Review History

Decision letter (RSOS-191553.R0)

22-Nov-2019

Dear Dr Ditlevsen:

It is a pleasure to accept your manuscript entitled "Distinguishing between parallel and serial processing in visual attention from neurobiological data" in its current form for publication in Royal Society Open Science.

Please ensure that you send to the editorial office an editable version of your accepted manuscript, and individual files for each figure and table included in your manuscript. You can send these in a zip folder if more convenient; ensuring that your main manuscript file is provided as an editable .doc, .docx, or .tex file. Figures should be publication quality EPS or PDF files. Failure to provide these files may delay the processing of your proof.

You can expect to receive a proof of your article in the near future. Please contact the editorial office (opencscience_proofs@royalsociety.org) and the production office (opencscience@royalsociety.org) to let us know if you are likely to be away from e-mail contact -- if you are going to be away, please nominate a co-author (if available) to manage the proofing process, and ensure they are copied into your email to the journal.

Kind regards,

on behalf of Dr Mark Walton (Associate Editor) and Dr Essi Viding (Subject Editor).
